# Cytotoxicity towards Breast Cancer Cells of Pluronic F-127/Hyaluronic Acid Hydrogel Containing Nitric Oxide Donor and Silica Nanoparticles Loaded with Cisplatin

**DOI:** 10.3390/pharmaceutics14122837

**Published:** 2022-12-17

**Authors:** Bianca de Melo Santana, Joana Claudio Pieretti, Rafael Nunes Gomes, Giselle Cerchiaro, Amedea Barozzi Seabra

**Affiliations:** Center for Natural and Human Sciences, Federal University of ABC, Santo André 09210-580, Brazil

**Keywords:** nitric oxide, nanoparticles, hydrogel, cytotoxicity, drug delivery, breast cancer cells

## Abstract

The incorporation of both nitric oxide (NO) donor (S-nitrosoglutathione, GSNO) and silica nanoparticles loaded with cisplatin (SiO_2_@CisPt NPs) into a polymeric matrix represents a suitable approach to creating a drug-delivery system with sustained and localized drug release against tumor cells. Herein, we report the synthesis, characterization, and cytotoxicity evaluation of Pluronic F-127/hyaluronic acid hydrogel containing GSNO and SiO_2_@CisPt NPs against breast cancer cells. SiO_2_@CisPt NPs were successfully synthesized, revealing a spherical morphology with an average size of 158 ± 20 nm. Both GSNO and SiO_2_@CisPt NPs were incorporated into the thermoresponsive Pluronic/hyaluronic hydrogel for sustained and localized release of both NO and cisplatin. The kinetics of NO release from a hydrogel matrix revealed spontaneous and sustained release of NO at the millimolar range for 24 h. The MTT assay showed concentration-dependent cytotoxicity of the hydrogel. The combination of GSNO and SiO_2_@CisPt incorporated into a polymeric matrix decreased the cell viability 20% more than the hydrogel containing only GSNO or SiO_2_@CisPt. At 200 µg/mL, this combination led to a critical cell viability of 30%, indicating a synergistic effect between GSNO and SiO_2_@CisPt NPs in the hydrogel matrix, and, therefore, highlighting the potential application of this drug-delivery system in the field of biomedicine.

## 1. Introduction

Breast cancer is the most commonly diagnosed and the fifth leading cause of mortality among different types of cancer, reaching over 2.3 million cases and over 680,000 deaths in 2021 [1]. Currently, chemotherapy is still the key treatment for breast cancer, employing drugs, such as taxanes (i.e., paclitaxel), doxorubicin, and platinum-based compounds (i.e., cisplatin) [2]. Another strategy is based on surgical resection, although there is high local cancer recurrence, which negatively impacts the prognosis and survival rates, requiring the use of adjuvant chemotherapy to reduce the relapse risk. The currently employed chemotherapeutics demonstrate high toxicity and systemic side effects, even when combined at different proportions to achieve more acceptable toxic profiles [3]. In this sense, further research is needed, focusing on more efficient treatments, which will minimize toxic effects.

Among different strategies, hydrogels consist of a promising platform for local and sustained therapy. The gels are hydrophilic and are usually composed of natural or synthetic biocompatible polymers [4]. More interestingly, hydrogels have intrinsic physicochemical and mechanical properties that provide room for encapsulating different agents, such as chemotherapeutics and, more recently, nanomaterials, thus promoting the controlled and sustained release of these agents directly to the target site of application [5]. Concerning breast cancer therapy, hydrogel matrices can be used for both topical and injectable treatments, as they promote tissue adhesion and, when injected, they can attain any shape, especially in circumstances of breast tumor resection.

Herein, we explore a Pluronics F-127/hyaluronic acid hydrogel combined with nanoparticles (NPs) and a nitric oxide (NO) donor. Pluronics are non-ionic surfactants, composed of polyethylene oxide-polypropylene oxide-polyethylene oxide (PEO-PPO-PEO) triblock copolymers. They are biocompatible bioadhesives, which have chemical stability and the ability to form micelles at low concentrations and/or temperatures, as well as the ability to undergo a sol–gel transition with the increase in concentration and/or temperature, leading to a hydrogel matrix at room or physiological temperature [6]. Pluronics can be used as a gel for topical applications or even as a thermosensitive injectable smart hydrogel for controlled drug delivery, since they allow controlled loading and release from the gel network. They can be injected as a polymer solution and then pass through a first-order transition, producing flexible soft hydrogel matrices in situ at the physiological temperature [7]. Due to their amphipathic capacity being above a certain concentration, which denominated the critical micellar concentration (CMC), the monomers can self-organize in an aqueous solution. The micellization process starts with an increase in temperature and/or concentration, with the segregation of polar segments (ethylene oxide units), which face the water, from nonpolar segments (propylene oxide units), which face the inner micelle part [8,9]. Under suitable conditions and with the increase in temperature (which can be adjusted to the room or physiological temperature), the micelles assume a packaging arrangement, leading to hydrogel formation [10]. Due to the thermal properties of Pluronics^®^, the substances of interest can be easily incorporated into the polymeric solution under low temperatures, such as NPs and NO donors, and with the increase in temperature (room temperature) the hydrogel containing the active drugs and nanoparticles is obtained [11]. In addition to that, hyaluronic acid (HA), a biopolymer formed by glucuronic acid and *N*-acetylglucosamine, can be used to improve the targetability of the formulation, since CD44, the HA receptor in cells, is overexpressed in breast cancer cells, such as the MDA-MB-231 [12,13].

Several nanomaterials have been used in nanomedicine. Among them, silica nanoparticles (SiO_2_ NPs) are interesting due to their porosity, hydrophilic nature, and easy-low-cost preparation, in addition to excellent biocompatibility, large surface area, and the possibility of controlling their morphology [14]. SiO_2_ NPs are considered non-toxic NPs, being very safe for use in the sustained release of drugs. Thus, the encapsulation of molecules in SiO_2_ NPs increases the drug delivery and absorption, in addition to reducing the toxicity that the loaded drug would have if it were freely circulating in the body [14]. Hence, we synthesized SiO_2_ NPs and loaded it with cisplatin (CisPt), a commonly used chemotherapeutic, to promote sustained delivery and reduced toxicity of the chosen drug. 

In addition to promoting more efficient drug delivery by incorporating CisPt-loaded SiO_2_ NPs into the Pluronic/hyaluronic acid hydrogel matrix, we aimed to combine our system with an NO donor. NO became globally known after being reported as a vasodilator, resulting in the Nobel Prize in Physiology or Medicine in 1998 [15]. NO plays pivotal roles as an endogenous signaling agent, being involved in diverse biological mechanisms. Regarding cancer therapy, the combination of NO and common chemotherapeutics is an increasingly reported tendency in the literature, mainly because NO has been shown to be an important sensitizer of cancer cells, improving the chemotherapeutic efficiency [16]. NO actuates in cancerous cells through several mechanisms, such as down-regulating P-glycoprotein, inhibiting NF-κB, and, more interestingly, depleting glutathione, which inactivates chemotherapeutic drugs, such as cisplatin. The incorporation of an NO donor (herein, S-nitrosoglutathione (GSNO)) into the hydrogel consists of a clever strategy to promote sustained NO release. GSNO acts as a spontaneous NO donor, due to the homogeneous cleavage of S-NO, releasing free NO. The superior viscosity of the hydrogel matrix favors the radical pair recombination of NO• and GS•, restoring the GSNO molecule and decreasing the rates of NO release [6].

In this report, we develop a Pluronics^®^/hyaluronic acid hydrogel containing CisPt-loaded SiO_2_ NPs and GSNO as the NO donor. SiO_2_ NPs were successfully synthesized and loaded with CisPt (loading efficiency of 58.06 ± 2.9%), showing a hydrodynamic size of 318 nm. The final formulation was homogeneous and promoted sustained and controlled release of NO over 24 h. Cytotoxicity assays were carried out against the triple-negative epithelial breast cancer adenocarcinoma MDA-MB-231 cell line, evidencing a synergistic effect from the combination of CisPt-loaded SiO_2_ NPs (SiO_2_@CisPt) and NO incorporated into a hydrogel matrix, with a pronounced reduction in cell viability when compared to control treatments. To the best of our knowledge, this is the first report to demonstrate the synergistic toxicity of GSNO and SiO_2_@CisPt into a hydrogel matrix against breast cancer cells.

## 2. Materials and Methods

### 2.1. Materials

Tetraethyl orthosilicate (TEOS), cisplatin, sodium nitrite, hydrochloric acid, reduced L-glutathione, phosphate-buffered solution (PBS), Pluronic-F127 [poly(ethylene oxide)-poly(propylene oxide)-poly(ethylene oxide)], hyaluronic acid sodium salt, Dulbecco’s Modified Eagle High Glucose Medium (DMEM), fetal bovine serum (FBS), trypsin-EDTA, penicillin, streptomycin, MTT reagent [3-(4,5-dimethylthiazol-2-yl)-2,5-diphenyltetrazolium bromide], and dimethyl sulfoxide (DMSO) were purchased from Sigma-Aldrich, St Louis, MO, USA. Ethanol was obtained from Synth, Diadema, SP, Brazil. All experiments were carried out using analytical grade water from the Millipore Milli-Q Gradient filtration system (Millipore, 18.2 MΩ).

### 2.2. Synthesis of SiO_2_ NPs

SiO_2_ NPs were prepared by the sol–gel method [17]. Briefly, 250 μL of the TEOS solution was diluted in 2.5 mL of water and added to 5 mL of ethanol/ammonium hydroxide (10:1 *v*/*v*) solution. The final mixture was stirred for 2 h to ensure that the reaction was complete. The obtained SiO_2_ NPs were isolated by centrifugation, washed with deionized water, and freeze-dried.

### 2.3. Synthesis of SiO_2_@CisPt NPs and Evaluation of Cisplatin Encapsulation Efficiency

SiO_2_ NPs were loaded with cisplatin using methods adapted from the literature [18,19]. SiO_2_ NPs were dispersed in water at a concentration of 1 mg·mL^−1^ and separately mixed with cisplatin (1:1 *w*/*w*). The materials were kept under stirring for 24 h, centrifuged, washed, and lyophilized. The cisplatin encapsulation efficiency was evaluated by inductively coupled plasma mass spectrometry (ICP-MS, (Agilent 7900, Hachioji, Japan), by quantifying the Pt concentration in the supernatant. The supernatant containing the remaining chemotherapeutic concentration (free cisplatin) was separated from the NPs by centrifugation, digested with double-distilled HNO_3_ at 100 °C for 1 h, diluted, and then analyzed. The loading efficiency was calculated according to Equation (1).
(1)$$\mathrm{Loading}\;\mathrm{efficiency}\;\left(\%\right)=\frac{\left(\mathrm{Initial}\;\mathrm{concentration}\;-\;\mathrm{Supernatant}\;\mathrm{concentration}\right)}{\left(\mathrm{Initial}\;\mathrm{concentration}\right)}\times 100$$

### 2.4. Synthesis of GSNO

The NO donor, GSNO, was synthesized by reacting reduced L-glutathione (1.23 mol·L^−1^) with an equimolar amount of sodium nitrite (NaNO_2_) in aqueous HCl solution (1.0 mol·L^−1^) for 40 min in an ice bath, protected from light. The obtained GSNO was precipitated by adding acetone, washed with cold water, vacuum filtered, and freeze-dried for 48 h. Solid GSNO was maintained in a desiccator at −20 °C. GSNO was characterized using UV–Vis spectrophotometry [11].

### 2.5. Preparation of Pluronic F-127 Hydrogels

All hydrogels were prepared with 25% (*w*/*v*) Pluronic F-127 and 0.05% (*w*/*v*) hyaluronic acid, as previously described with adaptations [6,20]. To this end, 1.25 g of solid Pluronic F-127 (PL) was added to 5.0 mL of PBS, kept at 10 °C overnight, leading to the complete dissolution of the PL. Next, 2.5 mg of hyaluronic acid (HA) was added to the PL solution in an ice bath, followed by gentle homogenization. This procedure led to the preparation of PL (25% *w*/*v*)/HA (0.05%) hydrogels, henceforth PL-HA.

For PL-HA hydrogels containing SiO_2_ NPs or SiO_2_@CisPt, 25 mg of NPs were added to the previously prepared PL-HA hydrogel at 5 °C and under gentle magnetic stirring. This process led to the formation of a PL-HA hydrogel containing 0.5 wt % of NPs, henceforth referred to as PL-HA-SiO_2_ or PL-HA-SiO_2_@CisPt. For PL-HA hydrogels containing GSNO, 42 mg of GSNO were added to the previously prepared PL-HA hydrogel at 5 °C and under gentle magnetic stirring. This process led to the formation of PL-HA hydrogel containing 25 mmol·L^−1^ of GSNO, henceforth referred to as PL-HA-GSNO. For hydrogels containing GSNO and SiO_2_ NPs or SiO_2_@CisPt NPs, a similar methodology was employed. The amounts of 42 mg of GSNO and 25 mg of NPs (SiO_2_ or SiO_2_@CisPt) were added to the previously prepared PL-HA, following the procedure described above. This process led to the formation of PL-HA hydrogel containing 25 mmol·L^−1^ of GSNO and 0.5 wt % of NPs, henceforth referred to as PL-HA-SiO_2_-GSNO or PL-HA-SiO_2_@CisPt-GSNO.

### 2.6. Characterization of the Prepared Materials

#### 2.6.1. X-ray Diffraction (XRD)

The structure of SiO_2_ NPs was investigated using XRD measurements. The analysis was performed in a STADI-P diffractometer (Stoe, Darmstadt, Germany) at room temperature, 50 kV, 40 mA, and using MoKα (λ = 0.7093 Å) radiation. A Mythen 1 K (Dectris, Baden, Switzerland) detector was used to collect X-ray photons of a powdered sample, in the 2θ range, from 5.0° to 64.265°, with step sizes of 0.0158°, and a counting time of 100 s at each occurrence at 0.785°.

#### 2.6.2. Fourier-Transformed Infrared Spectroscopy (FTIR)

FTIR analyses were recorded for SiO_2_@CisPt NPs using a Spectrum Two FTIR spectrometer (PerkinElmer, Waltham, MA, USA), equipped with ZnSe crystal for ATR mode, with a nominal angle of incidence of 45°. The analyses were recorded in the range of 700 to 4000 cm^−1^ at a resolution of 4 cm^−1^, employing a powdered sample.

#### 2.6.3. Dynamic Light Scattering (DLS)

The average hydrodynamic size, polydispersity index (PDI), and zeta potential (mV) of SiO_2_ NPs and SiO_2_@CisPt NPs were analyzed using a Zetasizer Nano ZS (Malvern Instruments Co, UK). The measurements were performed in aqueous media at 25 °C, using a disposable folded capillary zeta cell (10 mm path length) at a fixed angle of 173°.

#### 2.6.4. Scanning Electron Microscopy (SEM)

The morphology of lyophilized samples of SiO_2_@CisPt NPs and PL-HA-GSNO-SiO_2_@CisPt hydrogel was examined with a field emission scanning electron microscope Quanta 250 (FEI Co., Hillsboro, OR, USA), equipped with an Oxford X-MAX50 Energy Dispersive Spectrometer (EDS) (Oxford, UK), operated at 20 kV with a STEM (scanning transmission electron microscopy)-type detector. The dried samples were individually spread on double-sided conductive tape, placed in a metal stub, and then analyzed. The EDS technique was used to qualitatively map the distribution of carbon (C), oxygen (O), platinum (Pt), silicon (Si), and sulfur (S) atoms.

### 2.7. Kinetics of NO Release from GSNO-Containing PL

The kinetics of NO release from GSNO-containing hydrogel (PL-GSNO) was monitored by measuring changes in absorbance intensity at 545 nm (nN→π* transition), using a UV–visible spectrophotometer (Agilent 8454, Palo Alto, CA, USA) following GSNO decomposition. Changes in the absorbance intensity at 545 nm were associated with S–N bond cleavage and free NO release [6]. Kinetic data were acquired at 37 °C, for 23 h, in 1 h intervals. The initial GSNO concentration was 25 mmol·L^−1^. The NO-released concentration was calculated, according to Lambert–Beer Law (Equation (2)) and the results were reported as the mean ± standard deviation (SD) of two independent experiments and expressed in terms of NO concentration:[NO]_t_ = [GSNO]_0_ − [GSNO]_t_ = (A_0_b/ε_GSNO_) − (A_t_b/ε_GSNO_)(2)

Equation (2) relates NO concentration at time, t, ([NO]_t_) to the GSNO absorption band, where [GSNO]_0_ and [GSNO]_t_ are the concentrations of GSNO at the beginning of the reaction and at time, t, respectively. Variables A_0_ and A_t_ are the GSNO absorbances, at 545 nm, at the beginning of the monitoring, and at time, t, respectively. ε_GSNO_ is the molar absorption coefficient of GSNO at 545 nm (ε = 18.4 mol^−1^·L·cm^−1^), and b is the optical path of the cuvette (b = 1 cm). Moreover, initial rates of NO release through GSNO decomposition were determined using linear regression from the slopes of the initial sections of the kinetic curve [21].

### 2.8. In Vitro Diffusion of GSNO from PL-HA-SiO_2_@CisPt-GSNO

The in vitro diffusion of GSNO from PL-HA-GSNO-SiO_2_@CisPt was performed with a 7 mL vertical Franz diffusion cell (Hanson Research Corporation, CA). The cell consists of donor and receptor chambers that were separated by a hydrophilic nitrocellulose membrane with a 50 nm porosity and 25 mm diameter (Merck Millipore Ltd., County Cork, Ireland). The donor chamber was filled with 2 mL of PL-HA-SiO_2_@CisPt-GSNO (0.5 wt % of SiO_2_@CisPt NPs and 50 mmol·L^−1^ of GSNO). A volume of 7 mL of PBS, pH 7.4, at 37 °C, was added to the receptor chamber, with constant stirring. Every 30 min, a volume of 500 μL was withdrawn from the receptor chamber and replaced by the same volume of PBS. For the quantification of GSNO diffused from the hydrogels, the withdrawn samples were analyzed by UV-Vis spectrophotometry at λ = 336 nm.

### 2.9. Mathematical Models

To investigate the mechanism of GSNO diffusion from the hydrogels, the obtained kinetic curves of GSNO diffusion were adjusted through linear regression for the Higuchi, Hixson–Crowell, and Korsmeyer–Peppas mathematical models (Table 1). The release mechanism was determined by analyzing the correlation coefficient (R^2^). Where t represents the time; Q_t_, the cumulative amount of drug released at time, t; Q_0_, the initial cumulative amount of GSNO; Q_t_/Q_∞_, the fraction of drug released at time, t; K_K_, K_S_, and K_K_, the release constants for Higuchi, Hixson–Crowell, and Korsmeyer–Peppas, respectively; *n*, the release exponent for the Korsmeyer–Peppas model; *n* = 0.5 represents the Fickian diffusion; and a value of *n* > 0.5 represents anomalous (non-Fickian) diffusion.

### 2.10. Cell Culture and Cell Viability Assays

The cytotoxicity of PL-HA, PL-HA-SiO_2_, PL-HA-GSNO, PL-HA-SiO_2-_GSNO, PL-HA-SiO_2_@CisPt, and PL-HA-SiO_2_@CisPt-GSNO were evaluated against the triple-negative epithelial breast cancer adenocarcinoma MDA-MB-231 cell line (Federal University of ABC, Brazil). The cells were cultured in DMEM medium, supplemented with 10% (*v*/*v*) fetal bovine serum and antibiotic containing 10,000 µL of penicillin and 10 mg.L^−1^ of streptomycin, at a temperature of 37 °C in a controlled atmosphere composed of 5% CO_2_ (Cell Culture Greenhouse—Forma Series II, Hepa Class 200, Thermo Scientific). Culture flasks with 80% cell growth confluence were used for assays. To perform plating, cells were washed with PBS, trypsinized, and subjected to centrifugation (1500 rpm for 5 min). The supernatant was discarded, and the pellet resuspended in 5 mL of DMEM medium. After resuspension, 10 µL aliquots of cell suspension from this solution were subjected to counting in a Neubauer chamber for use in the cell viability assays.

The effects of the PL-based hydrogels on the viability of MDA-MB-231 cells were evaluated by MTT assay. The cells were cultured and plated in 96-well plates at a density of 4.104 cells·cm^−2^ and exposed to treatments with PL-HA, PL-HA-SiO_2_, PL-HA-GSNO, PL-HA-SiO_2_-GSNO, PL-HA-SiO_2_@CisPt, and PL-HA-SiO_2_@CisPt-GSNO at different concentrations, for 24 h, as shown in Table 2. After the treatment, 30 µL of a 5 mg·mL^−1^ MTT solution was added directly to the medium that already contained the treatments and incubated at 37 °C for 1 h, protected from light. After this, the culture medium was completely removed and 150 µL of DMSO was added for incubation with agitation and protected from light for 15 min. The samples were then read on a microplate reader (CELER Polaris) at a wavelength of 570 nm [22]. The assays for each group were performed in quadruplicate. The percentage of viable cells was calculated using the absorbance of the negative control as 100% viability. Significant statistical difference (*p*-value < 0.05) between the treatments and control were calculated using ANOVA one-way.

## 3. Results and Discussion

### 3.1. Synthesis and Characterization of SiO_2_@CisPt NPs

In this work, SiO_2_ NPs were synthesized by the sol–gel method and loaded with CisPt. The encapsulation efficiency of Cis into SiO_2_ NPs was found to be 58.06 ± 2.9%, with a final CisPt concentration of 576.48 ± 23.40 µg per mg of nanoparticle. According to previous reports, CisPt loading efficiency is typically between 2% and 70%, with final concentrations of around 100 µg of chemotherapeutic agent per mg of NP [23,24,25]. The loading efficiency and final concentration of CisPt into SiO_2_ NPs reported in this work are within the expected range.

Figure 1a shows the XRD pattern for the SiO_2_ NPs. A broad peak centered at 2θ = 21.5° is attributed to the existence of an SiO_2_ amorphous structure, as previously reported [17,26], confirming the formation of SiO_2_ NPs. Figure 1b shows the FTIR spectrum of SiO_2_@CisPt NPs. The sharp peaks at 1052.6 cm^−1^ and 949.3 cm^−1^ are related to asymmetric and symmetric Si-O-Si stretching vibrations of SiO_2_ NPs, respectively [17,27]. The peak at 3220.6 cm^−1^ is related to the OH of SiO_2_ NPs [17]. No peak related to platinum was found, possibly because the concentration was too low to be detected by the equipment. 

DLS analyses (Table 3) showed that the average hydrodynamic size, polydispersity index (PDI), and zeta potential for SiO_2_ NPs were 197.53 ± 6.0 nm, 0.144 ± 0.044, and −27.37 ± 0.51 mV, respectively. For SiO_2_@CisPt NPs, the values were 317.9 ± 2.6 nm (hydrodynamic size), 0.353 ± 0.04 (PDI), and −20.2 ± 0.75 mV (zeta potential). After incorporating cisplatin into SiO_2_ NPs, it is possible to notice an increase in the hydrodynamic size. Similar results have been reported for SiO_2_-based NPs [28,29]. The results indicate that both particles are nanoscale-sized with moderate-to-good dispersion, display homogeneous size distribution in aqueous media, and have a negative surface charge due to the presence of hydroxyl groups in SiO_2_ NPs [30].

The morphology of SiO_2_@CisPt NPs was characterized by SEM. Figure 2 shows the SEM microscopy (Figure 2a) of SiO_2_@CisPt NPs and the corresponding size distribution (Figure 2b). As can be observed, SiO_2_@CisPt NPs have uniform spherical shape morphology and are well-distributed, with an average size of 158 ± 20 nm for the solid state. These results are in accordance with other previously reported microscopies of SiO_2_-based NPs [31,32].

### 3.2. Synthesis and Characterization of Pluronic-F127-Based Hydrogels

The morphology of hydrogel was observed by SEM analysis. Figure 3 shows representative SEM images of the PL-HA-SiO_2_@CisPt-GSNO hydrogel, revealing a soft surface with the presence of interconnected pores. These pores might favor drug diffusion and delivery, as they can facilitate the diffusion of active drugs/nanomaterials.

In addition, EDS was employed for the identification of the elemental composition of the material. Figure 4 shows the PL-HA-GSNO-SiO_2_@CisPt hydrogel mapping for oxygen (Figure 4a), carbon (Figure 4b), sulfur (Figure 4c), platinum (Figure 4d), and silicon (Figure 4e) atoms. We can observe a high density of carbon and oxygen atoms, due to the Pluronic matrix, and a homogeneous distribution of sulfur, silicon, and platinum among the polymeric matrix, with the absence of microdomains.

### 3.3. Kinetics of NO Release from GSNO-Containing PL

GSH is a precursor molecule of GSNO, the NO donor. For GSNO synthesis, GSH was nitrosated from the reaction with an equimolar amount of sodium nitrate (NaNO_2_) in acidic medium (Equation (3)).
GSH + HNO_2_ → GSNO + H_2_O(3)

In this reaction, nitrous acid (HNO_2_) plays the role of the nitrosating agent, which is formed by the dissolution of nitrite (NO_2_^−^) in acidified aqueous solution. GSNO undergoes a spontaneous decomposition, yielding NO and oxidized glutathione (GSSG), as shown in Equation (4).
2 GSNO → 2 NO + GSSG(4)

After synthesis, GSNO was incorporated into PL-HA-GSNO hydrogel. The kinetic profile of NO release from the PL-HA-GSNO hydrogel is shown in Figure 5. The NO release profile from the hydrogel showed that GSNO can release NO through the polymer matrix at 37 °C for at least 23 h. The concentration of the NO released is in the order of mmol·L^−1^. Therapeutic effects were reported in the literature when NO was administered in this order of concentration, such as antitumor and antimicrobial actions [33,34].

The initial rate of NO release from the hydrogel was 1.09 ± 0.24 mmol·L^−1^·h^−1^ and the maximum concentration of NO released from the PL-HA-GSNO hydrogel was approximately 18.5 mmol·L^−1^, after 23 h of monitoring. This value is lower when compared to the maximum NO released from GSNO dissolved in water reported in other pieces of work, which was found to be 36 mmol L^−1^ [6]. This can be explained by the higher viscosity of the matrix of hydrogel (compared to water), which promotes the cage effect favoring the recombination of the radical pair of GS• and NO•, restoring the GSNO molecule, prolonging the stability of the GSNO, and consequently the NO release rate. Thus, it is possible to control the medium viscosity to modulate the rate of GSNO decomposition and, consequently, the NO release [6,11]. Importantly, Figure 5 shows that sustained NO release at the millimolar range is obtained by the incorporation of GSNO into the hydrogel, making this biomaterial suitable for biomedical applications.

### 3.4. In Vitro Diffusion of GSNO from PL-HA-SiO_2_@CisPt-GSNO

The diffusion of GSNO from PL-HA-SiO_2_@CisPt-GSNO was monitored by the vertical Franz diffusion cell, for 6 h at 37 °C. The amount of GSNO diffused from the hydrogel at the donor chamber through the membrane to the receptor chamber was quantified by the detection of the characteristic absorption band of GSNO at 545 nm (ε = 18.4 L.mol^−1^·cm^−1^). Figure 6 shows the in vitro diffusion of GSNO at physiological pH. The initial diffusion rate of GSNO was found to be 17.98 ± 1.83 mmol·L^−1^. Furthermore, 40% of the GSNO incorporated in the hydrogel was diffused in 1 h, at pH 7.4.

Comparing the diffusion profile of GSNO from the hydrogel observed in this work (Figure 6) with the total amount of GSNO diffused from Pluronic F-127 hydrogel reported by us previously, it is possible to notice that in the present work, there was a higher diffusion rate of GSNO [6]. This difference can be attributed to the types of membranes applied in each assay, as well as the temperature. The temperature is directly proportional to diffusion rates, since higher temperatures increase the energy and movement of the molecules, increasing the diffusion rates, whereas the type of membrane can repulse or attract the molecules, facilitating or not the diffusion. Although Pelegrino et al. performed the assay at 32.5 °C using a polysulfone membrane, which is hydrophobic, our assay was performed at 37 °C using a nitrocellulose membrane, which is hydrophilic, and consequently favors the GSNO diffusion from the hydrogel [6].

Diffusion mechanisms of GSNO from the hydrogel were investigated by applying Higuchi, Hixson–Crowell, and Korsmeyer–Peppas mathematical models to the data obtained in Figure 6. Table 4 shows the correlation coefficients (R^2^) for linearized data obtained by the mathematical models. According to the literature, the Higuchi mathematical model indicates a classic Fickian diffusion of an active compound, whereas the Hixson–Crowell involves matrix erosion. Therefore, the Korsmeyer–Peppas model is characterized by the diffusional constant “n”. For n ≤ 0.45, the main mechanism controlling the release of the drug into the system is a classic Fickian diffusion; when n ≥ 0.89, the equation corresponds to zero-order release kinetics (non-Fickian diffusion); and when 0.45 < n < 0.89, it indicates anomalous transport kinetics—a combination of case II diffusion and matrix relaxation or release by erosion [21] (Urzedo et al., 2020).

The most suitable mathematical model for GSNO diffusion was Korsmeyer–Peppas, with a correlation coefficient of 0.978 and K_K_ constant of 39.11%.h^−n^. The diffusional constant n was found to be 0.936, indicating an anomalous transport, where we have a case-II transport controlled by the relaxation of the matrix, plus a Fickian diffusion [35] (Cellet et al., 2015). Similar studies reported diffusion of S-nitrosothiols, mainly governed by Fickian diffusion [6,21,36].

### 3.5. In Vitro Cytotoxicity of the Hydrogels against MDA-MB-231 Tumor Cells

The cell viability assay was carried out using MDA-MB-231 tumor cells. This cell line of breast cancer epithelial adenocarcinoma is highly aggressive and poorly differentiated (ECACC). MDA-MB-231 cells were treated for 24 h with six different groups of hydrogels: PL-HA, PL-HA-SiO_2_, PL-HA-GSNO, PL-HA-SiO_2_-GSNO, PL-HA-SiO_2_@CisPt, and PL-HA-SiO_2_@CisPt-GSNO. The correlation between each hydrogel component and its concentration in the treatments is listed in Table 2 (Section 2.10), and the cell viability results can be seen in Figure 7.

For cells treated with PL-HA and PL-HA-SiO_2_ (Figure 7a,b, respectively) there was no toxic effect within the range of concentrations tested, since all the cell viabilities remained above 80%, meeting the criteria of ISO 10993-5 to be considered as non-cytotoxic. This corroborates that the hydrogel matrix and empty SiO_2_ NPs are biocompatible. For PL-HA-GSNO treatment (Figure 7c), toxic effects were only noticed at 500 and 1000 µmol·L^−1^ of GSNO (according to Table 2), which resulted in viability percentages of 65 and 50%. The results are in accordance with previous publications in which higher GSNO concentrations can cause cell toxicity [33].

Figure 7d shows that for PL-HA-SiO_2_-GSNO (empty SiO_2_ NPs), the addition of SiO_2_ NPs did not alter the hydrogel toxicity, since the decrease in cell viability was very similar to PL-HA-GSNO at 500 and 1000 µmol·L^−1^ of GSNO. Hydrogels without GSNO and containing CisPt-loaded SiO_2_ NPs (PL-HA-SiO_2_@CisPt, Figure 7e) did not showed a relevant decrease in cell viability up to 50 µg·mL^−1^ of NPs, whereas a significantly decrease in cell viability was observed for hydrogels containing 100 and 200 µg·mL^−1^ of SiO_2_ NPs loaded with CisPt, resulting in cell viabilities of 64 and 53%, respectively. This indicates that hydrogels containing higher amounts of NPs loaded with CisPt have a toxic effect against MDA-MB-231 cells, but the cell viability values are very similar to the hydrogel containing just GSNO. Finally, hydrogels containing both GSNO and CisPt-loaded SiO_2_ NPs demonstrated a higher toxic effect towards MDA-MB-231 cells. As can be seen in Figure 7f, PL-HA-SiO_2_@CisPt-GSNO has a higher toxic effect against cancer cells at 5000 and 10,000 µg·mL^−1^ of hydrogel (which corresponds to 500 and 1000 µmol·L^−1^ of GSNO; 100 and 200 µg·mL^−1^ of SiO_2_ NPs loaded with CisPt). Indeed, PL- PL-HA-SiO_2_@CisPt-GSNO at 5000 and 10,000 µg·mL^−1^ of hydrogel concentration decreased cell viability to 46 and 29%, respectively.

Taken together, the hydrogel matrix (unloaded or loaded with empty SiO_2_ NPs) was not toxic to cancer cells, as expected, since both the hydrogel and SiO_2_ NPs are known to be biocompatible. Whereas hydrogels containing either GSNO or CisPt-loaded SiO_2_ NPs were toxic at higher tested concentrations. The incorporation of SiO_2_@CisPt NPs into the hydrogel matrix had a significant cytotoxic effect when compared with the hydrogel containing empty SiO_2_ NPs (which are a naturally inert nanomaterials) due to the chemotherapeutic action of CisPt. Importantly, the combination of the NO donor (GSNO) and CisPt-loaded SiO_2_ NPs in the hydrogel matrix further decreased cancer cell viability at the higher tested concentrations. Interestingly, these results indicate that NO donor and CisPt-loaded SiO_2_ can be more effective in decreasing breast cancer cell viability if applied in combination.

CisPt is a polychemotherapeutic agent used to treat various types of cancer, such as breast, ovarian, head, neck, etc., which kills cancer cells through DNA damage, inhibition of DNA synthesis and mitosis, and induction of apoptotic cell death [37]. Cisplatin is capable of generating reactive oxygen species (ROS) through the activation of a family of enzymes named nicotinamide adenine dinucleotide phosphate oxidase (NOX). They are capable to convert NADPH into NADP^+^, releasing electrons, which are accepted by O_2_ molecules, generating superoxide (O_2_^•−^). Superoxides can be processed by superoxide dismutase (SOD) and ultimately generate ROS, which can cause anticancer effects and also severe side effects, such as cardiac toxicity and nephrotoxicity of cisplatin [38]. The IC50 values of MDA-MB-231 cells treated with free CisPt for 24 h are reported to be in the range of 19 to >250 µmol/L [39,40,41,42]. According to Figure 7e the concentrations of PL-HA-SiO_2_@CisPt that significantly decreased cell viability were 100 and 200 µg/mL of NPs, which correspond to 191 and 382 µmol/L of CisPt. These values are in the range of the values reported in the literature. It should be noted that in our work, CisPt was encapsulated into SiO_2_ NPs and further incorporated into PL-HA, which imposes a physical barrier for the free diffusion of the drug, allowing a sustained CisPt releasing.

At higher concentrations (micro-mili molar range), NO can also cause oxidative stress, inducing cell death. Different reactive oxygen and nitrogen species (ROS and RNS, respectively) can be generated from NO depending on the redox, acidic, and hypoxic conditions of the cell. In addition to NO, ROS and RNS can lead to significant post-translational modifications, including nitration, nitrosylation, and oxidation of important biomolecules involved in cell death and proliferation [43]. Under pathological conditions, NO reacts with O_2_^•−^ producing the oxidizing agent peroxynitrite (ONOO^−^). This leads to oxidative DNA damage, cytochrome C release, increased mitochondrial permeability, lipid peroxidation, and protein nitration and oxidation [44]. These reactions trigger cellular responses ranging from modulations involving cell signaling to oxidative damage, affecting cells by necrosis or apoptosis [16]. Chu and collaborators developed a cocktail of polyprodrug nanoparticles fabricated from branched polyprodrug amphiphiles of NO and cisplatin and demonstrated that in situ released NO could react with the cisplatin-activated generation of O_2_^•−^ to further pump out highly reactive ONOO^−^ [45]. Additionally, a NO-releasing Pt(IV) prodrug, Pt–furoxan, can release cisplatin and NO, which can modulate the cellular response towards cisplatin, leading to a synergistic anti-proliferation and anti-metastasis effect both in vitro and in vivo [46].

Moreover, NO donors can have a direct toxic effect towards cancer cells or can trigger cancer cell sensitization to the traditional chemotherapeutic drugs, such as CisPt [47]. Indeed, the combination of NO donors with nano/biomaterials can be applied alone to promote toxic effects on cancer cells, or, more recently, in combination with chemotherapy or radiotherapy agents [16,48]. The latter approach is a cooperative strategy, in which due to the vasodilation effect of NO, in in vivo applications, NO can enhance tumor permeability, facilitating nanoparticle-carrying anti-cancer drug accumulation at the tumor site [49]. Finally, NO-releasing biomaterial, in combination with CisPt-loaded NPs can promote sustained NO and CisPt release directly to the target site of application (i.e., tumor site) rather than circulating to other tissues/organs at sufficiently high concentrations for the therapeutic effect, avoiding side effects. Further studies are required to better investigate the cell death mechanism of both NO donors and in cooperation with CisPt-loaded SiO_2_ NPs.

## 4. Conclusions

We designed a PL hydrogel containing both GSNO and SiO_2_@CisPt NPs to promote combined treatment and sustained release of both the chemotherapeutics and NO. The chemotherapeutic-containing SiO_2_ NPs were successfully synthesized and incorporated into the PL matrix. Similarly, GSNO was incorporated into the hydrogel and promoted sustained NO release for 24 h. In short, the combination of SiO_2_@CisPt with GSNO, in a hydrogel matrix, resulted in a synergistic effect, leading to higher cytotoxicity against MDA-MB-231 cells, which may result from the activation of different cell death pathways, being an efficient and promising strategy for the treatment of resistant cancer lineages. Therefore, the developed formulation may find important applications in the field of cancer treatment due to the synergistic effect promoted by combining SiO_2_@CisPt and GSNO, in addition to the versatility of PL-hydrogel applications for localized applications.

## Figures and Tables

**Figure 1 pharmaceutics-14-02837-f001:**
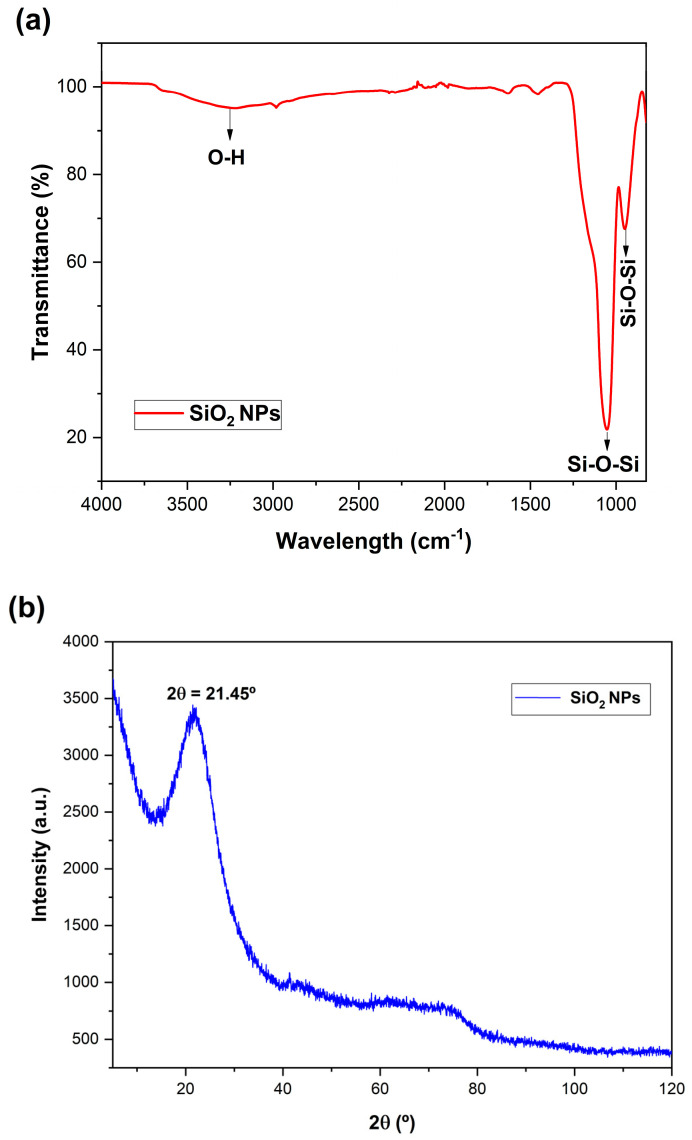
FTIR spectrum of SiO_2_@CisPt NPs (**a**). XRD pattern of SiO_2_ NPs synthesized by the sol–gel method (**b**).

**Figure 2 pharmaceutics-14-02837-f002:**
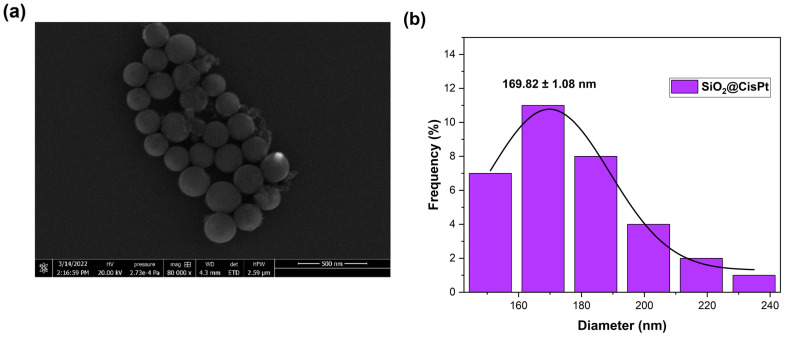
Representative SEM image of SiO_2_@CisPt NPs (**a**) and the corresponding size distribution (**b**) of SiO_2_@CisPt NPs.

**Figure 3 pharmaceutics-14-02837-f003:**
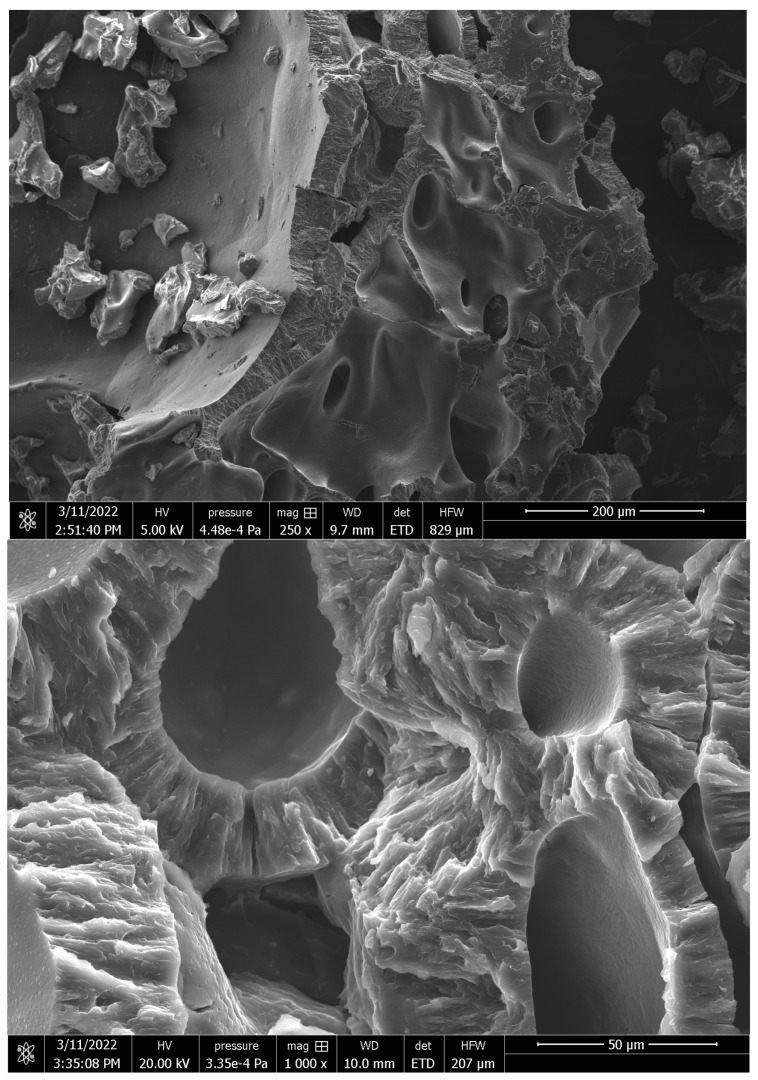
Representative SEM images of PL-HA-GSNO-SiO_2_ @CisPt lyophilized hydrogel.

**Figure 4 pharmaceutics-14-02837-f004:**
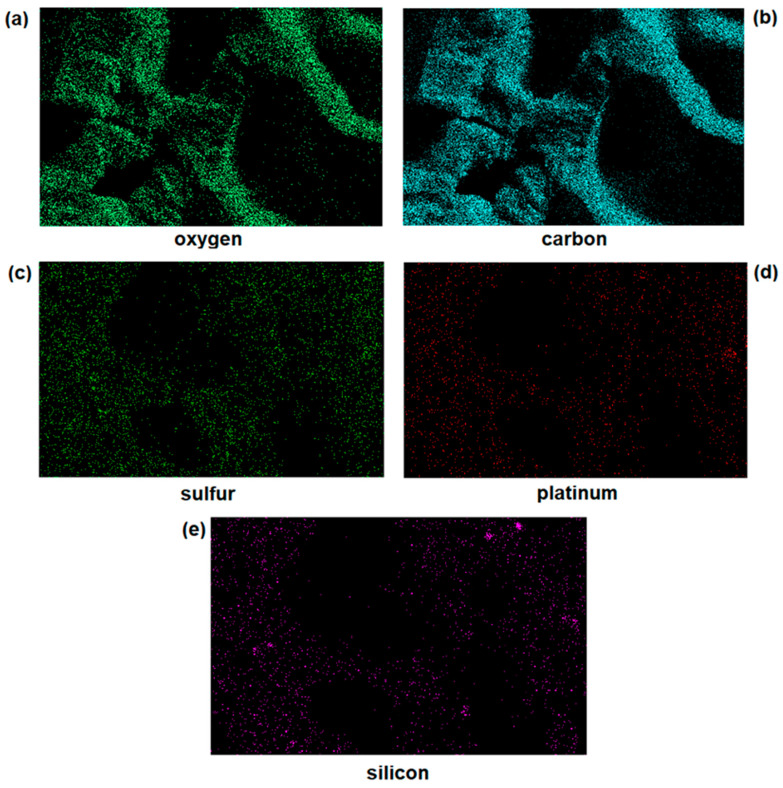
EDS mapping of oxygen (**a**), carbon (**b**), sulfur (**c**), platinum (**d**), and silicon (**e**) atoms of PL-HA-GSNO-SiO_2_@CisPt lyophilized hydrogel.

**Figure 5 pharmaceutics-14-02837-f005:**
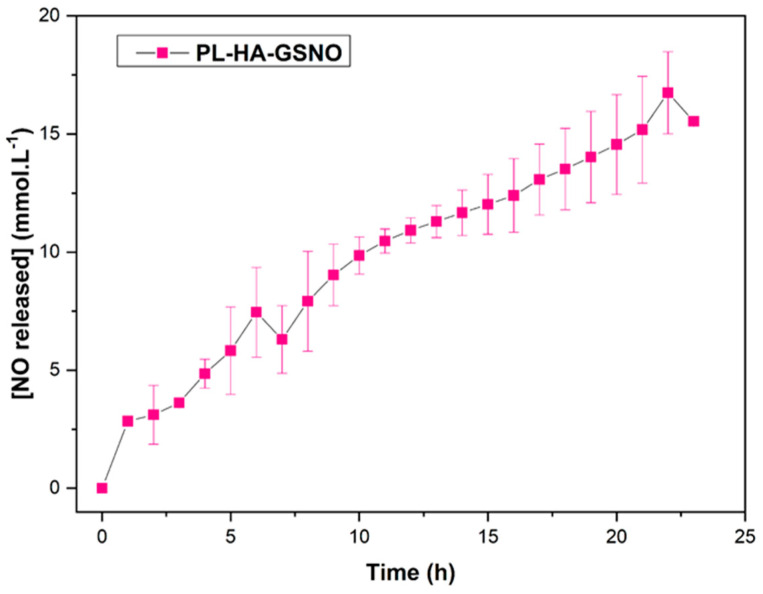
Kinetics of NO release from PL-HA-GSNO hydrogel (initial GSNO concentration 25 mmol·L^−1^) at 37 °C, monitored for 24 h. The results are reported as the mean ± standard deviation of two independent experiments.

**Figure 6 pharmaceutics-14-02837-f006:**
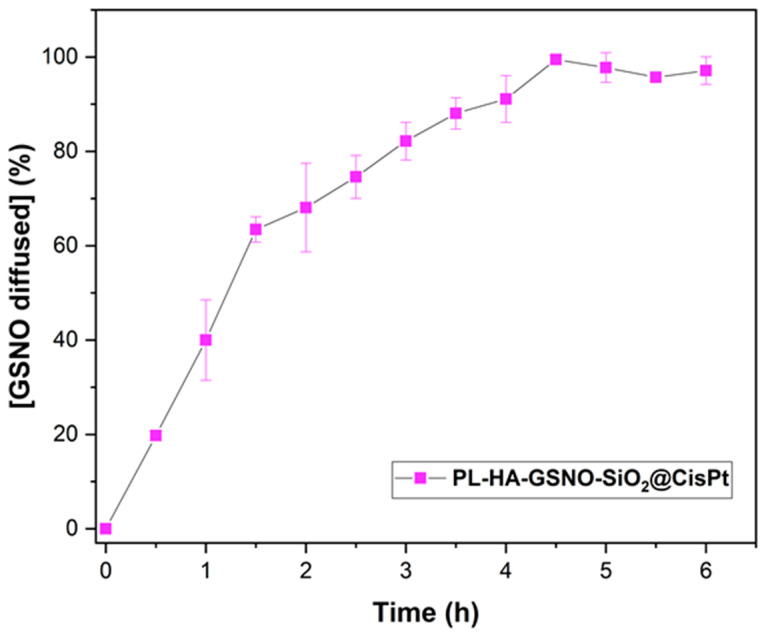
Diffusion profile of GSNO from the PL-HA-SiO_2_@CisPt, monitored for 6 h, at 37 °C. The results are reported as the mean ± standard deviation of three independent experiments.

**Figure 7 pharmaceutics-14-02837-f007:**
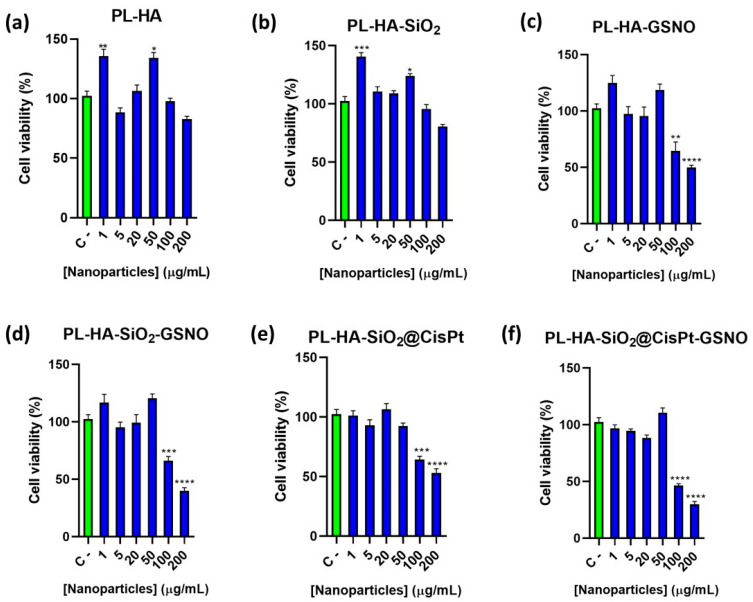
Viability of MDA-MB-231 cells treated with different concentrations of PL-HA (**a**), PL-HA-SiO_2_ (**b**), PL-HA-GSNO (**c**), PL-HA-SiO_2_-GSNO (**d**), PL-HA-SiO_2_@CisPt (**e**), and PL-HA-SiO_2_@CisPt-GSNO (**f**). Green columns indicate the control group and blue columns indicates the treatments. Error bars indicate the standard error (SE) and asterisks indicate significant difference (*p* < 0.05) between the treatments and control, calculated by ANOVA one-way. * *p* < 0.05, ** *p* < 0.02, *** *p* < 0.0007, and **** *p* < 0.0001.

**Table 1 pharmaceutics-14-02837-t001:** Mathematical models used for adjusting the diffusion data obtained from a vertical Franz diffusion cell.

Model	Equation
Higuchi	Q_t_= K_H_ · t^0.5^
Hixson–Crowell	Q_0_^1/3^ − Q_t_^1/3^ = K_S_ · t
Korsmeyer–Peppas	Ln $\frac{Q_{t}}{Q_{\infty }}$ = ln K_K_ + *n* · ln t

**Table 2 pharmaceutics-14-02837-t002:** Corresponding concentrations for Pluronic F-127 hydrogel, hyaluronic acid (HA), GSNO, SiO_2_ NPs, and CisPt used on the MDA-MB-231 cell line for MTT assay.

Pluronic F-127 (µg·mL^−1^)	Hyaluronic Acid (µg·mL^−1^)	GSNO (µmol·L^−1^)	SiO_2_ NPs (µg·mL^−1^)	CisPt (µmol·L^−1^)
5	0.125	5	1	1.900
250	0.625	25	5	9.560
1000	2.50	100	20	38.36
2000	6.25	250	50	95.65
5000	12.5	500	100	191.5
10,000	25.0	1000	200	382.9

**Table 3 pharmaceutics-14-02837-t003:** Hydrodynamic diameter, polydispersity index (PDI), and zeta potential values for SiO_2_ NPs and SiO_2_@CisPt NPs in water.

DLS Parameter	SiO_2_ NPs	SiO_2_@CisPt NPs
Hydrodynamic diameter (nm)	197.53 ± 6.0	317.9 ± 2.6
Polydispersity index (PDI)	0.144 ± 0.044	0.353 ± 0.04
Zeta potential (mV)	−27.37 ± 0.51	−20.2 ± 0.75

**Table 4 pharmaceutics-14-02837-t004:** Release parameters for GSNO diffusion from PL-HA-GSNO-SiO_2_@CisPt NPs according to Higuchi, Hixson–Crowell, and Korsmeyer–Peppas mathematical models.

Mathematical Models						
Higuchi		Hixson–Crowell		Korsmeyer–Peppas		
R^2^	K_H_ (%·h^−1/2^)	R^2^	K_s_ (%·h^−1^)	R^2^	K_K_ (%·h^−n^)	n
0.969	30.81	0.925	10.45	0.978	39.11	*0.936*

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
