# Peer review of "Cytotoxicity towards Breast Cancer Cells of Pluronic F-127/Hyaluronic Acid Hydrogel Containing Nitric Oxide Donor and Silica Nanoparticles Loaded with Cisplatin"

_pharmaceutics, 2022, doi:10.3390/pharmaceutics14122837_

Round 1

Reviewer 1 Report

In the study, the authors report the synthesis, characterization, and cytotoxicity evaluation of Pluronic F-127/hyaluronic acid hydrogel containing GSNO and SiO2@CisPt NPs against breast cancer cells. This manuscript should be accepted after minor revision. Some comments:

1.       The topic of manuscript is unclear and should be revised to highlight the content of paper.

2.       The total pages of 18 is much long for reader. The author had better condense or put some to sopport materials.

3.       Section 2.5, why the author added 0.05% (w/v) hyaluronic acid?

4.       The authors use UV for NO release in place of Gress method. UV method is simple, but it is not sentive for low NO release.

5.       The number of significant digits in Table 2 is not scientific. The authors should check the whole manuscript.

6.       The authors can use normal cells to test side effect and blood for hemolysis assay in futher study.

Author Response

Please, see attached file.

Reviewer 2 Report

The paper entitled “Cytotoxicity of nitric oxide releasing Pluronic F-127 hydrogel/silica nanoparticles loaded with cisplatin towards breast cancer cells” describes the synthesis, characterization, and evaluation of the cytotoxic potential of a drug-delivery system comprised of hydrogel matrix containing GSNO, as a NO donor, and/or silica nanoparticles loaded with cisplatin against triple-negative breast cancer cell line MDA-MB-231. This study aimed to develop a suitable drug-delivery system with the sustained and localized release of NO and cisplatin against mentioned breast adenocarcinoma cells. There are extensive literature data about the efficacy of different drug-delivery systems which have been developed to minimize toxic effects and potentiate the anticancer activity of the loaded drug itself. So far, many researchers have reported great antitumor activity of chemotherapeutics such as cisplatin loaded into different types of carriers against diverse cancer cells and possible molecular mechanisms of this agent have been widely described and published. Nevertheless, I dare to conclude that this paper is bringing an average level of novelty to the scientific community and is contributing to the literature.

Even though I do not consider that more study-based explanations are required for confirming the results, several questions need to be addressed and some weaknesses from the technical point of view need to be amended.

The manuscript is overall well-written, easy to read, and generally presented in a well-structured manner. In general, the study is well conducted and the results are more or less well presented. The cited references are relevant to the study and are mostly recent publications, though there is a certain number of self-citations. However, I do have some questions and suggestions for the authors about the methodology and results concerning cytotoxicity evaluation and would appreciate it if the authors can give short explanations.

Q1: In Section 2.10. (Line 249),  the authors state thatAfter the treatment, 200 μL of a 5 mg.mL-1 MTT solution was added directly to the medium and incubated for 1 h, protected from light.” How the volume of 200 μL of MTT solution was added to the medium in wells when the total volume of each well in the 96-well plate is a maximum of 200 μL? Did you mean that MTT was dissolved in medium forming MTT solution which was added directly to the adherent cells in wells after the treatment removal? If so, please correct the main text.

Q2: In Section 2.10. (Line 252), the authors state thatThe samples were then read on a microplate reader (CELER Polaris) at a wavelength of 570 nm”. Did the authors measure the absorbance of the samples at a wavelength of 670 nm for background correction? If so, please mention that in the main text.

Q3: In Section 2.10., which statistical test was used for data processing? After the sentence “The assays for each group were performed in quadruplicate” (Line 254) please indicate that the significance of the differences between various treatments was determined by a suitable statistical test (name which one). Also, write which values of p were considered significant.

Q4: In the Section 2.10. (Line 254) was written “The percentage of viable cells was calculated using the absorbance of the negative control as 100 % viability”. Why the authors didn’t insert negative control in the graphics in Figure 7? I strongly recommend inserting the negative control which will depict 100 % cell viability as the first column on both graphics in Figure 7, as this will improve the interpretation of the results shown in the figure and make it easier to understand.

Q5: I noticed some incorrectness in Figure 7 which the authors should explain. Even though the authors claimed that “Figure 7a shows that PL-HA and PL-HA-SiO2 NPs did not significantly decrease the cell viability for any tested concentrations” (Line 396), it is noticeable that there is some statistical significance for the certain concentrations that is not highlighted as it should be. For example, in Figure 7a, PL-HA in concentrations 250 µg.mL-1 (corresponds to 25 µmol.L-1 of GSNO and 5 µg.mL-1 of SiO2 NPs) and 10000 µg.mL-1 (corresponds to 1000 µmol.L-1 of GSNO and 200 µg.mL-1 of SiO2 NPs) did decrease cell viability, as well as PL-HA-SiO2 NPs  in concentration 200 µg.mL-1 of SiO2 NPs. The first two are not recognized as significant and are missing asterisks while the third one is properly marked in Figure 7a but is not in concordance with the mentioned statement in the main text. Furthermore, PL-HA in the concentration of 5000 µg.mL-1 (corresponds to 500 µmol.L-1 of GSNO and 100 µg.mL-1 of SiO2 NPs) had almost the same impact on cell viability as PL-HA-SiO2 NPs in concentration 100 µg.mL-1 and yet only PL-HA-SiO2 NPs has been recognized as significant, even though I dare say that both values are statistically insignificant. The same applies to one higher concentration where the substances had almost the same impact on cell viability, but only one is marked as significant though it seems both of them are indeed statistically significant. Similarly, in Figure 7b, PL-HA-SiO2-GSNO in concentrations 5000 and 10000 µg.mL-1 (corresponds to 500/1000 µmol.L-1 of GSNO and 100/200 µg.mL-1 of SiO2 NPs) did significantly decrease cell viability but the asterisk is missing.

If the authors emphasize exactly in relation to what statistical significance refers to and what is compared with what, maybe the claims and statements will be clearer. The authors are kindly asked to provide proper explanations, add asterisks in Figure 7 above appropriate columns, and revise/adjust the statements to improve the Discussion part of the manuscript.

The authors should also address the following issues in the Discussion section. In order to improve the quality of the manuscript, I advise you to supplement the text in the Discussion section with a few references indicating the IC50 values of cisplatin in the tested breast cancer cell line MDA-MB-231 after 24 h treatment. Additionally, discuss why the concentrations of cisPt incorporated into PL-HA-SiO2 NPs and PL-HA-SiO2-GSNO that are considered efficient are ranging from 190 to 380 µmol.L-1 which is so much higher than when used without a drug-delivery system. Isn’t it the purpose to minimize the dose of chemotherapeutics used for therapy in order to reduce side effects and potential resistance of cancer to chemotherapy? Please support with references the well-known mechanism of action of cisPt to induce the reactive oxygen species-mediated apoptosis and discuss how that can be related to synergistic action between cisPt and GSNO observed in this study.

Furthermore, the authors are kindly asked to correct formal inconsistencies in order to further ameliorate the quality of the manuscript, such as coordinated/consistent usage of the abbreviations throughout the whole manuscript (Introduction, M&M section, figures, tables, etc.), correctly stating the numbers of equations, tables, and figures, etc.

In some instances, the manuscript was not prepared in line with the journal’s guidelines, for example, the References list. As it is stated in the Instructions for Authors, in the text of the manuscript reference numbers should be placed in square brackets [ ], and placed before the punctuation; for example [1], [1–3], or [1,3]. Please correct.

Author Contributions and Conflicts of Interest are missing. Please add those sections.

As it is stated in the Instructions for Authors, Acronyms/Abbreviations/Initialisms should be defined the first time they appear in each of three sections: the abstract; the main text; the first figure or table. All tables and figures have a caption that should provide explanations of the abbreviations (all abbreviations within the table/figure must be defined when mentioned for the first time), and information on the applied statistical procedures. Please correct all the tables/figures.

Please make consistency between the figure itself, figure caption, and explanations in the main text (meaning: if it is a in the figure itself, then figure caption and main text should also state a and not A). Please correct throughout the whole manuscript and figures.

Please arrange all Figures, in the sense of aligning pictures or graphics within figures.

Please correct the following:

Line 12: Replace the word “tumoral” with “tumor” or “cancer” cells. Tumoral is an adjective and is used for describing the response, growth, potential, etc. but the cell itself is a tumor or cancer cell.

Line 16: Correct ‘”SiO2@ NPs” to “SiO2@CisPt NPs”.

Line 20: The sentence “The combination of GSNO and SiO2@CisPt into a polymeric matrix halved cell viability compared to the hydrogel containing only GSNO or SiO2@CisPt” is not entirely correct because it didn’t make it twice as small as the cellular viability of the hydrogel containing only GSNO or SiO2@CisPt but rather this combination further reduced it. Please correct the word halved with a more correct one. Also, insert the word “incorporated” before the “…into a polymeric matrix.”

Line 24: Replace “biomedical field” with “field of biomedicine”.

Line 38: Replace “with more reasonable toxic effects” with “which will minimize toxic effects”.

Line 44: Insert the word “thus” before “…promoting controlled and sustained…”.

Line 45: Replace the word “applications” with “therapy”.

Line 49: “Pluronics® F-127 hydrogel” is written here with the symbol “®” but not throughout the whole manuscript. Please make consistency throughout the whole manuscript when mentioning it and write it everywhere with or without the symbol “®”. The same applies to the Title and Abstract.

Line 59: Remove the hyphen from the phrase “in-situ”.

Line 80: Correct the abbreviation for cisplatin from “Cis” to “CisPt”. Also correct in Lines 82, 99, 101, 105, 132, 258, and 259. Please make sure that the abbreviations are consistent throughout the whole manuscript (Introduction, M&M section, figures, figure titles and captions, tables, etc.).

Line 80: Remove the word “drug” after the word “chemotherapeutic”.

Line 85: Replace “Nobel Prize from Medicine and Physiology” with “Nobel Prize in Physiology or Medicine”.

Line 87: Replace the word “applications” with “therapy”.

Line 89: Replace “sensitizer agent” with just “sensitizer” or  ”sensitizing agent”.

Line 108: Insert the word “incorporated” before the “…into a hydrogel matrix.”

Line 126: First time mentioning the abbreviation SiO2@CisPt NPs in the main text without explanation. All abbreviations must be defined when mentioned for the first time so I suggest introducing this abbreviation in the Introduction section and leaving it here like this.

Line 201: Correctly state the numbers of equations. Please replace “Equation (1)” with “Equation (2)”.

Line 226: In the part of the text “KK, KS, and KK” use a subscript for indicated letters, as written in the equations in the corresponding table. Also, correct the same in Line 377.

Lines 233 and 234: Inconsistency in abbreviations PL-HA-GSNO-SiO2 and PL-HA-GSNO-SiO2@CisPt”. Please correct to correspond to the ones introduced in Section 2.5. Also, correct in Lines 296, 297, 301, 303, 310, 374, 386, 387, 404, 415, 418, 422, in Figure 6, and Figure 7. Please make sure that the abbreviations are consistent throughout the whole manuscript (Introduction, M&M section, figures, figure titles and captions, tables, etc.).

Line 240: Insert the word “viability” before the “assays.”

Line 249: The content of Table 2 does not correspond to what was written in the text. The content of Table 4 does. Please correct and reorganize the numbering of all tables. Make sure that the numbers of tables are consistent throughout the whole manuscript and correspond to the statements in the main text.

Lines 265 and 268: Disparity in statements in the main text and Figure 1 caption. As described in Figure 1 caption, Fig. 1a is the FTIR spectrum and 1b is the XRD pattern. Please correct either in manuscript text or Figure 1 caption so that they correspond to each other. Also, please make consistency between the figure itself, figure caption, and explanations in the main text (meaning: if it is a in the figure itself, then figure caption and main text should also state a and not A). Please correct throughout the whole manuscript and figures.

Lines 276 and 285: Please, correct the numbering of the tables (replace “Table 2” with “Table 3”). Also, correct table numbering in Lines 365 and 374. Please correct and reorganize the numbering of all tables. Make sure that the numbers of tables are consistent throughout the whole manuscript and correspond to the statements in the main text.

Line 328: Replace “when compared to the” with “than the”.

Line 362: Correct “PL-HA-SiO2@CisPt” to “PL-HA-SiO2@CisPt-GSNO”. At the end of the Figure 6 caption, insert how the results are presented, like in Figure 5 (where it is written “as the mean ± standard deviation of two independent experiments”).

Lines 365 and 377: In “R2” and “h-n” use a superscript for indicated letters, as written in the corresponding table.

Lines 388, 393, and 394:  Please, correct the numbering of the table (replace “Table 4” with “Table 2”).

Line 423: Please indicate compared to what or which sample all given p values are considered significant (in relation to what statistical significance refers to and what is compared with what). The certain p value indicates significant differences between various treatments so it must be clearly stated between which treatments comparisons are made. 

Line 449: Remove a full stop after the word “apoptosis” and before the reference.

Line 453: Replace “toxic effects to cancer cells” with “toxic effects on cancer cells”.

Taken together, I am at the statement that this paper is acceptable for publication but in a revised form. 

Author Response

Please, see attached file.
